# Supervised learning with incomplete data via sparse representations

## Abstract

This paper addresses the problem of training a classifier on a dataset with missing features and its application to a complete or incomplete test dataset. A supervised learning method is developed to train a general classifier, such as a logistic regression or a deep neural network, using only a limited number of observed entries (features), assuming sparse representations of data vectors on an unknown dictionary. The pattern of missing entries is independent of the sample and can be random or structured. The proposed method simultaneously learns the classifier, the dictionary and the corresponding sparse representations of each input data sample. A theoretical analysis is also provided comparing this method with the standard imputation approach, which consists on performing data completion followed by training the classifier based on their reconstructions. The limitations of this last "sequential" approach are identified, and a description of how the proposed new "simultaneous" method can overcome the problem of indiscernible observations is provided. Additionally, it is shown that, if it is possible to train a classifier on incomplete observations so that its reconstructions are well separated by a hyperplane, then the same classifier also correctly separates the original (unobserved) data samples. Extensive simulation results are presented on synthetic and well-known reference datasets that demonstrate the effectiveness of the proposed method compared to traditional data imputation methods.

## 1 Introduction

In many machine learning applications sometimes the measurements are noisy or affected by artifacts, resulting in incomplete data samples. Examples of this situation include: self-driving vehicle or robot where objects in the view field can be partially occluded; recommendation systems built from the information gathered by different users where not all the users have fully completed their forms; or medical datasets where typically not all tests can be performed on all patients.

A classical approach of supervised learning with missing entries is to apply entry imputation as a preprocessing step, followed by standard training based on these reconstructions (Little & Rubin, 2014). For example, some techniques compute missing entries using statistical methods, such as the "mean", "regression" and "multiple" imputation methods (Little & Rubin, 2014). Other completion methods are based on machine learning ideas by estimating missing entries through $K$-nearest neighbor (Batista & Monard, 2002), Self Organization Maps (SOM) (Fessant & Midenet, 2002), multilayer or recurrent neural networks (Yoon & Lee, 1999; Bengio & Gingras, 1995), tensor completion algorithms (Solé-Casals et al., 2018) and others (García-Laencina et al., 2009). In (Huang et al., 2018), a technique for completing the matrix of features was proposed by imposing low-rankness and incorporating the label information into the completion process. A different

approach is to avoid direct imputation of lost inputs and rely on a probabilistic model of input data, learning model parameters through the Expectation Maximization (EM) algorithm and building a Bayesian classification (Ghahramani & Jordan, 1993; Little & Rubin, 2014). However, the latter "model-based" approach has the disadvantage that it requires a good probabilistic data model, which is usually not available, especially for real-world applications such as those involving natural images.

On the other hand, during the last few years in the signal processing community, there has been a rapid development of the Compressed Sensing (CS) theory and sparse coding algorithms that exploit the redundancy of natural signals (Candes et al., 2006; Donoho, 2006; Eldar & Kutyniok, 2012; Gregor & LeCun, 2010). A particular case of CS is the signal completion problem, i.e. the reconstruction of signals from incomplete measurements. Several algorithms for matrix and tensor (multidimensional signals) completion have been recently proposed relying on different low-dimensional manifold models that are fit to the available entries, allowing for an extrapolation of the unobserved entries. This category of models ranges from sparse representation of vectors (Aharon et al., 2006; Fadili et al., 2008) and tensors (Caiafa & Cichocki, 2012) to low-rank matrix (Candes & Tao, 2010) and tensor decompositions (Liu et al., 2012; Yuan et al., 2018). The success of sparse representations in many signal processing applications motivated researchers to use dictionary learning techniques for data classification (Mairal et al., 2008), either using class-specific dictionaries (Ramirez et al., 2010; Sprechmann & Sapiro, 2010), or using a single one for all classes (Tosic & Frossard, 2011).

In this paper, the goal is to develop a method for training a classifier using incomplete data samples with their labels and to identify the conditions under which such a classifier performs as good as the ideal classifier, i.e. the one that could be obtained from complete observations. We propose to use the sparse representation model for data vectors, and analyze the limitations of the sequential approach, i.e. imputation followed by training (section 2). In section 3 we introduce the simultaneous classification and coding approach, which consists of incorporating a sparse data representation model into a cost function optimized for training the classifier and, at the same time, finding the best sparse representation of the observed data. We provide theoretical conditions under which we can guarantee that the obtained classifier is as good as the ideal classifier. In section 4, a computational method is presented to train a classifier on incomplete data; in section 5, extensive experimental results are presented using synthetic and well known benchmark datasets illustrating the effectiveness of our proposed algorithms; and finally, in section 6, the main conclusions are outlined.

## 1.1 PRELIMINARY DEFINITIONS AND PROBLEM FORMULATION

We assume a supervised learning scenario with vector samples and their labels $\{\mathbf{x}_i, y_i\}$, $i = 1, 2, \ldots I$, $\mathbf{x}_i \in \mathbf{R}^N$ and $y_i \in \{0, 1, \ldots, C - 1\}$ ($C$ classes), so that we only have access to subsets of entries in each data vector along with their labels, which is denoted by $\{\tilde{\mathbf{x}}_i, y_i\}$, $\tilde{\mathbf{x}}_i \in \mathbf{R}^M$ with $M < N$.

The success of CS theory is based on the fact that sparse coding of natural signals is nearly ubiquitous. It is found, for example, in the way that neurons encode sensory information (Olshausen & Field, 1996; 1997; Lee et al., 2007). We define the set of all $K$-sparse vectors $\Sigma_K^L = \{\mathbf{s} \in \mathbf{R}^L$ such that $\|\mathbf{s}\|_0 \leq K\}$ (containing at most $K$ non-zero entries) and assume that data vectors $\mathbf{x}_i$ admit $K$-sparse representations over an unknown dictionary $\mathbf{D} \in \mathbf{R}^{N \times L}$ ($L \geq N$), i.e. $\mathbf{x}_i = \mathbf{D}\mathbf{s}_i$, with $\mathbf{s}_i \in \Sigma_K^L$. The columns $\mathbf{d}_i \in \mathbf{R}^N$ of a dictionary are called "atoms" because we can express any data vector as a combination of at most $K$ atoms.

Without losing generality, to simplify the mathematical treatment, we shuffle the entries of the full vector $\mathbf{x}_i$ to have the observed values placed in the upper $M$ positions, so that we can write:

$$\mathbf{x}_i = \begin{bmatrix} \tilde{\mathbf{x}}_i \\ \mathbf{z}_i \end{bmatrix}, \text{ with } \mathbf{z}_i \in \mathbf{R}^{(N-M)}. \tag{1}$$

However, our results are valid for any pattern of missing entries for each data vector. Accordingly, we define $\mathbf{D}_i \in \mathbf{R}^{N \times L}$ as the shuffled version of $\mathbf{D}$, such that it stacks in the top $M$ rows those corresponding to the observed entries:

$$\mathbf{D}_i = \begin{bmatrix} \tilde{\mathbf{D}}_i \\ \mathbf{E}_i \end{bmatrix}, \text{ with } \tilde{\mathbf{D}}_i \in \mathbf{R}^{M \times L} \text{ and } \mathbf{E}_i \in \mathbf{R}^{(N-M) \times L}. \tag{2}$$

Some important uniqueness properties of sparse vector reconstructions are characterized by the Restricted Isometry Property (RIP) of a matrix, which is defined as follows: a matrix $\mathbf{A} \in \mathbf{R}^{N \times L}$ satisfies the RIP of order $K$ if there exists $\delta_K \in [0, 1)$ such that $(1 - \delta_K)\|\mathbf{s}\|_2^2 \leq \|\mathbf{A}\mathbf{s}\|_2^2 \leq (1 + \delta_K)\|\mathbf{s}\|_2^2$, holds for all $\mathbf{s} \in \Sigma_K^L$. RIP was introduced in the context of CS by Candes and Tao (Candès & Tao, 2005) and characterizes matrices which are nearly orthonormal when operating on sparse vectors.

Let us assume that a perfect classifier $p_\Theta(y \mid \mathbf{x})$ in a two classes scenario ($y_i \in \{0, 1\}$), e.g. a logistic regression or deep neural network, can be trained on the complete dataset $\{\mathbf{x}_i, y_i\}$, such that $0.5 < p_\Theta(y_i = 1 \mid \mathbf{x}_i) \leq 1.0$ and $0 \leq p_\Theta(y_i = 0 \mid \mathbf{x}_i) \leq 0.5$, $\forall i = 1, 2, \ldots, I$, where $\Theta$ is the set of trained parameters. We want to develop a method to estimate the classifier parameters $\hat{\Theta}$ using only the incomplete data $\{\tilde{\mathbf{x}}_i, y_i\}$ and to identify the conditions under which such classifier is compatible with the ideal classifier. In the following section we will show that, when data imputation based on sparsity followed by training is used, some limitations are clearly identified.

## 2 The problem of training after data imputation

First, we will provide some intuition about the limitation of the sparsity-based imputation or "sequential" approach through a toy example. We consider the classification of hand-written digit images belonging to two classes: the numbers "3" and "8". We will assume that they admit 2-sparse representations over a dictionary and only the right halves of these digits are observed. Let us consider two example vectors $\mathbf{x}_i$ and $\mathbf{x}_j$ belonging to classes "3" and "8", respectively, as shown in Figure 1 (a-d). Here, we are clearly faced with the *indiscernible observations* problem because our observations of two vectors from different classes are identical ($\tilde{\mathbf{x}}_i = \tilde{\mathbf{x}}_j$). It is obvious that at least two possible 2-sparse representations for the observed data exist, let's say $\tilde{\mathbf{x}}_i = \tilde{\mathbf{D}}_i\mathbf{s}_i = \tilde{\mathbf{D}}_i\mathbf{s}'_i$, with $\mathbf{s}_i \neq \mathbf{s}'_i$. Since the solution is not unique, we are not able to correctly reconstruct the original data vectors based only on the observations and the sparsity assumption.

If the $K$-sparse representation of the observations $\tilde{\mathbf{x}}_i$ are unique, then $\mathbf{x}_i$ can be perfectly reconstructed from the incomplete observations and the classifier can be successfully trained using these reconstructions, but unfortunately this is not the case most of the times. In the particular case where the dictionary $\mathbf{D}$ is known in advance, there are some conditions on the sampling patterns based on properties of matrices such as coherence, spark or RIP that can guarantee a correct reconstruction (Eldar & Kutyniok, 2012). However, these conditions are difficult to meet in practice. Moreover, in the more general case where the dictionary $\mathbf{D}$ is unknown and needs to be learned from data, it is even more difficult to obtain reconstructions of partially observed data vectors that are well separated. For example, even in the case where complete data vectors are well separated, it could happen that their reconstructions become difficult to separate, leading to suboptimal classifiers.

In this paper, motivated by the limitation of the sequential approach, we propose to incorporate the class information of incomplete samples in order to learn simultaneously their sparsest representations and the classifier, such that the classification error of the reconstructions is minimized.

## 3   The simultaneous classification and coding approach

In the previous section, we showed that sparsity-based imputation methods using only the information on available entries are prone to fail because the non-uniqueness of solutions can make the training of a good classifier an impossible task. It is interesting to note that we could solve this problem by incorporating from the beginning the information of the class to which the incomplete data vectors belong to. Let us consider the toy example of previous section, for which we assumed the classes are linearly separable and each data vector admits a $K$-sparse representation over a dictionary $\mathbf{D} \in \mathbf{R}^{N \times L}$. A two-dimensional simplified visualization for this example is provided in Figure 1(e). When we apply a sparsity-based reconstruction algorithm on observations $\tilde{\mathbf{x}}_i$, and $\tilde{\mathbf{x}}_j$ such that $\tilde{\mathbf{x}}_i = \tilde{\mathbf{x}}_j$ and, assuming for now that dictionary $\mathbf{D}$ is known, we search for the $K$-sparse representation that is compatible with the measurements, i.e. we find $\mathbf{s} \in \Sigma_K^L$ such that $\tilde{\mathbf{x}}_i = \mathbf{D}_i \mathbf{s}$. In this case, there are at least two equally acceptable solutions $\mathbf{s}_i$ and $\mathbf{s}_j$ corresponding to vectors $\mathbf{x}_i$ and $\mathbf{x}_j$, respectively. If we were to make the wrong decision of assigning $\mathbf{s}_j$ to $\mathbf{x}_i$ and $\mathbf{s}_i$ to $\mathbf{x}_j$, then the resulting set of reconstructed vectors would not be linearly separable, which is illustrated in Figure 1(f), therefore it would be impossible to find a linear classifier. The key observation here is that, in order to assign the proper sparse representation we could also look at the label information $y_i$ and $y_j$, and make the solution unique by discarding any $K$-sparse representation resulting in not separable vector reconstructions. This suggests that we would need to train the classifier and find the proper representation not only as sparse as possible but also providing the best separation of classes. In other words, we should simultaneously learn the optimal classifier and find sparse representation of observations. Next, we theoretically analyze the case of using a logistic regression classifier and demonstrate that, if we are able to train a classifier on incomplete observations such that their sparse representations are well separated by a hyperplane, then the same classifier correctly separates the original (unobserved) data vectors.

### 3.1   Theoretical guarantees in the logistic regression case

Let us consider the case of a logistic regression classifier (Hastie et al., 2009) where the set of parameters $\Theta = \{\mathbf{w}, b\}$ are a vector $\mathbf{w} \in \mathbf{R}^N$ and a scalar (bias) $b \in \mathbf{R}$, and a perfect classifier exists if there is a hyperplane that separates both classes, i.e., for each data vector $\mathbf{x}_i$: $f(\mathbf{x}_i) = \mathbf{w}^T \mathbf{x}_i + b > 0$ when $y_i = 1$ and $f(\mathbf{x}_i) \leq 0$ if $y_i = 0$. We investigate the conditions under which a perfect classifier can be trained using incomplete data samples and their labels. By using the same shuffling as in equation (1), we can write the partition of the shuffled coefficients $\mathbf{w}_i \in \mathbf{R}^N$ as follows:

$$\mathbf{w}_i = \begin{bmatrix} \tilde{\mathbf{w}}_i \\ \mathbf{u}_i \end{bmatrix}, \text{ where } \mathbf{u}_i \in \mathbf{R}^{(N-M)}. \tag{3}$$

Since the classifier is based on the feature $f(\mathbf{x}_i) = \mathbf{w}_i^T \mathbf{x}_i + b = \tilde{\mathbf{w}}_i^T \tilde{\mathbf{x}}_i + \mathbf{u}_i^T \mathbf{z}_i + b$, the vector $\mathbf{u}_i \in \mathbf{R}^{(N-M)}$ contains the weights associated with unobserved entries $\mathbf{z}_i$. The following definition introduces a way to measure the amount of discriminative information associated with missing entries.

**Definition 3.1. Discriminative Missing Information (DMI)** $\mu_i$: *Given a classifier $\{\mathbf{w}, b\}$ and a partition of observed/missing entries in sample vector $\mathbf{x}_i$ such that the shuffled vector of coefficients $\mathbf{w}_i$ can be partitioned as in eq. (3), we define the DMI for sample $\mathbf{x}_i$ as:* $\mu_i = \|\mathbf{u}_i\|_1 = \sum_{m=1}^{N-M} |\mathbf{u}_i(m)|$.

Intuitively, if the DMI is small enough, the classification will not strongly depend on the unobserved entries $\mathbf{z}_i$ and then a classifier could be trained from incomplete data samples. The following theorem establishes precise conditions under which the existence of a classifier obtained from incomplete

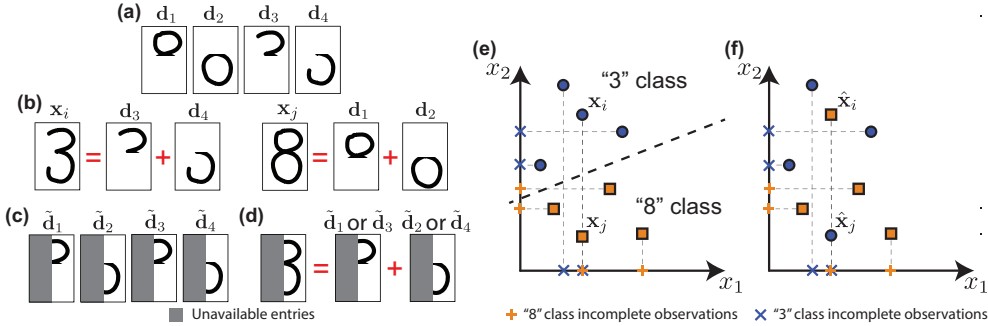

Figure 1: An *indiscernible observation* toy example: (a) 4 out $L$ dictionary elements $\mathbf{d}_i$ (atoms). (b) Digit "3" and "8" can be represented by combining only two atoms in the dictionary (2-sparse representations). (c) Atoms with their left-halves occluded $\tilde{\mathbf{d}}_i$. (d) An occluded digit "3" or "8" admits more than one 2-sparse representation, for example, they can be expressed as the sum of occluded atoms $\tilde{\mathbf{d}}_1$ and $\tilde{\mathbf{d}}_2$, or $\tilde{\mathbf{d}}_3$ and $\tilde{\mathbf{d}}_4$. (e) Simplified two-dimensional representation of samples $\mathbf{x}^T = [x_1, x_2] \in \mathbf{R}^2$ from "3" and "8" classes that are linearly separable where incomplete observations are taken by observing only one entry. Note that $\mathbf{x}_i$ and $\mathbf{x}_j$ belong to different classes but their observations are identical. (f) A wrong reconstruction of data vectors, i.e. $\hat{\mathbf{x}}_i \neq \mathbf{x}_i$ and $\hat{\mathbf{x}}_j \neq \mathbf{x}_j$ can make the set of reconstructed vectors not linearly separable.

data can be guaranteed and thus, a perfect classifier can be successfully trained. The proof can be found in Appendix A

**Theorem 3.2.** *Given a dataset $\{\mathbf{x}_i, y_i\}$, $i = 1, 2, \ldots, I$ with normalized data vectors ($\|\mathbf{x}_i\| \leq 1$) admitting a $K$-sparse representation over a dictionary $\mathbf{D} \in \mathbf{R}^{N \times L}$ with unit-norm columns and satisfying the RIP of order $K$ with constant $\delta_K$, and suppose that, we have obtained an alternative dictionary $\mathbf{D}' \in \mathbf{R}^{N \times L}$ also satisfying the RIP of order $K$ with constant $\delta_K$ such that, for the incomplete observation $\tilde{\mathbf{x}}_i \in \mathbf{R}^M$, the $K$-sparse representation solution is non-unique, i.e. $\exists \mathbf{s}_i, \mathbf{s}'_i \in \Sigma_K^L$ such that $\tilde{\mathbf{x}}_i = \tilde{\mathbf{D}}_i \mathbf{s}_i = \tilde{\mathbf{D}}'_i \mathbf{s}'_i$, where $\tilde{\mathbf{D}}_i, \tilde{\mathbf{D}}'_i \in \mathbf{R}^{M \times L}$ are the matrices containing the observed rows of $\mathbf{D}$ and $\mathbf{D}'$, respectively; $\mathbf{s}_i \in \mathbf{R}^L$ is the vector of coefficients of the true data, i.e. $\mathbf{x}_i = \mathbf{D}_i \mathbf{s}_i$ and $\mathbf{s}'_i$ provides a plausible reconstruction through $\hat{\mathbf{x}}_i = \mathbf{D}'_i \mathbf{s}'_i$ with $\|\hat{x}_i\| \leq 1$. If a perfect classifier $\{\mathbf{w}_i, b\}$ of the reconstruction $\hat{\mathbf{x}}_i$ exists such that*

$$\epsilon > 2K\mu_i / (1 - \delta_K), \tag{4}$$

*with $\epsilon > 0$ being the minimum distance of reconstructed data sample to the separating hyperplane, then the full data vector $\mathbf{x}_i$ is also perfectly separated with this classifier, in other words: $f(\mathbf{x}_i) = \mathbf{w}_i^T \mathbf{x}_i + b > 0$ ($\leq 0$) if $y_i = 1$ ($y_i = 0$).*

It is interesting to note that, supposing that we have obtained a classifier from incomplete data, if we are able to evaluate condition (4) then we can figure out whether such a classifier is optimal by testing the condition of the theorem. However, It is well known that, in general, the RIP constant is difficult to compute in practice. When the dictionary is highly uncorrelated[1], i.e. if $\rho K < 1$, then the RIP constant can be written in terms of the correlation coefficients $\rho$ as follows: $\delta_K = (K - 1)\rho$ (Eldar & Kutyniok, 2012), and $\rho$ is easy to compute. Our theorem provides useful insights of the problem. For example, we it suggests the following conditions are desirable: (a) very well separated reconstructed vectors (large $\epsilon$); (b) very sparse model (small $K$); (c) small norm variability of transformed $K$-sparse vectors through $\mathbf{D}$ (small $\delta_K$), which can be interpreted as a

---

[1]Correlation coefficient is the maximum absolute correlation between any two columns in a normalized dictionary.

quasi-orthonormal basis behaviour (low correlation, for example); and (d), small weights assigned to unobserved entries of $\mathbf{x}_i$ (small DMI $\mu_i$).

## 4 The proposed method

Here, we propose to combine the training of the classifier together with the learning of a dictionary and optimal sparse representations such that reconstructed data vectors are compatible with the observed entries and, at the same time, well separated. To do that we propose to minimize the following global cost function, where we consider that $\mathbf{x}$, $\hat{\mathbf{x}}$ and $\mathbf{m}$ are now the original unruffled vectors and the pattern of missing features is not the same for each data sample $i$ :

$$J(\Theta, \mathbf{D}, \mathbf{s}_i) = \frac{1}{I} \sum_{i=1}^{I} \big\{ J_0(\Theta, \hat{\mathbf{x}}_i, y_i) + \lambda_1 J_1(\mathbf{D}, \mathbf{s}_i) + \lambda_2 J_2(\mathbf{s}_i) \big\}, \tag{5}$$

where $\Theta$ contains the classifier parameters, i.e. the vector of coefficients and bias for a logistic regression model, or the vector of weights in a deep neural network classifier architecture; $\mathbf{D} \in \mathbf{R}^{N \times L}$ ($L \geq N$) is a dictionary and $\mathbf{s}_i \in R^L$ are the representation coefficients such that the reconstructed data vectors are $\hat{\mathbf{x}}_i = \mathbf{D}\mathbf{s}_i$. $J_0(\Theta, \hat{\mathbf{x}}_i, y_i)$ is a measure of the classification error for sample $i$. Typically, we use the crossentropy measure, i.e. $J_0(\Theta, \mathbf{x}, y_i) = -\log(p_{\Theta}^{y_i}(\mathbf{x}))$, where $p_{\Theta}^{y_i}(\mathbf{x}_i)$ is the probability assigned by the classifier to sample $\mathbf{x}_i$ as belonging to class $y_i$. $J_1(\mathbf{D}, \mathbf{s}_i)$ is a measure of the error associated with the sparse representation when it is restricted to the observed entries, and is defined as follows: $J_1(\mathbf{D}, \mathbf{s}_i) = \frac{M}{N} \|\mathbf{m}_i * (\mathbf{x}_i - \mathbf{D}\mathbf{s}_i)\|^2$, where $*$ stands for the entry-wise product, $\mathbf{m}_i \in \mathbf{R}^N$ is the observation mask for sample $i$, with $m_i(n) \in \{0, 1\}$ such that $m_i(n) = 1$ or $0$ if data entry $\mathbf{x}_i(n)$ is available or missing, respectively. $J_2(\mathbf{s}_i) = \frac{1}{N} \|\mathbf{s}_i\|_1$ is the $\ell_1$-norm whose minimization promotes the sparsity of the representation (Candès & Tao, 2005). Finally, the hyper-parameters $\lambda_1$ and $\lambda_2$ allow to give more or less importance to the representation accuracy and its sparsity, with respect to the classification error. Intuitively, minimizing equation (5) favors solutions that not only have sparse representations compatible with observed entries, but also provides reconstructions that are best separated in the given classes.

To minimize the cost function in equation (5) we propose to alternate between the optimization over $\{\Theta, \mathbf{D}\}$ and $\mathbf{s}_i$ ($i = 1, 2, \dots, I$). As usual, we adopt a first order (gradient based) search of minima, where gradients of functions $J_0(\cdot)$, $J_1(\cdot)$ and $J_2(\cdot)$ are easily derived. Note that in the case of feedforward architectures, the back-propagation technique can be used for the first term $J_0(\cdot)$. It is also noted that function $J_2(\cdot)$ is not differentiable at zero, so we need to avoid zero crossing in every update. In the case where the given test dataset is also incomplete we can use the dictionary learned during the training phase to find the sparsest representation for the given observations, compute the corresponding full vector reconstructions and apply the classifier to them. The algorithms are presented in Appendix B.

## 5 Experimental results

We synthetically generated $I = 11,000$ ($10,000$ training $+1,000$ test) $K$-sparse data vectors $\mathbf{x}_i \in \mathbf{R}^{100}$ using a dictionary $\mathbf{D} \in \mathbf{R}^{100 \times 200}$ obtained from a Gaussian distribution with normalized atoms, i.e. $\|\mathbf{D}(:, l)\| = 1, \forall l$. A random hyperplane $\{\mathbf{w}, b\}$ with $\mathbf{w} \in \mathbf{R}^N$, $b \in \mathbf{R}$ was randomly chosen dividing data vectors into two classes according to the sign of the expression $\mathbf{w}^T \mathbf{x}_i + b$, which defines the value of label $y_i$. We also controlled the degree of separation between the two classes by regenerating all data vectors with distances to the hyperplane lower than a pre-specified threshold $d$. For each case we generated 10 realizations using different masks and input data. We applied our simultaneous method, with hyperparameters $\lambda_1$ and $\lambda_2$ tuned via cross-validation, to train a

logistic regression classifier on the training dataset with randomly distributed missing entries, and compared the obtained Test Accuracy against the following standard sequential methods: **Seq. Sparse**: reconstructions are obtained by finding the sparsest representation compatible with the observations solving a LASSO problem; **Zero Fill (ZF)**: missing entries are filled with zeros, which is equivalent to ignore unknown values; **Mean Unsupervised (MU)**: missing entries are filled with the mean computed on the available values in the same position in the rest of data samples; **Mean Supervised (MS)**: as in the previous case but the mean is computed on the samples of the same class vectors only; **K-Nearest Neighbor (KNN)**: as in the previous case but the mean is computed on the K-Nearest Neighbors of the same class only. Since the objective here is to compare the performance of obtained classifiers, we computed the accuracy (mean $\pm$ standard error of the mean - s.e.m.) on the test dataset using all the methods for two cases of degree of separation between classes ($d = 0.0, 0.2$), two levels of sparsity ($K = 4, 32$) and missing values in the training dataset ranging from 25% to 95% as shown in Fig. 2. Our results show that the simultaneous algorithm clearly outperforms all the sequential methods. A t-test was performed to evaluate the statistical significance of the difference between our algorithm (Simul) and MS. It is interesting to note that, when classes has some degree of separation ($d = 0.2$), using simple methods as computing the mean or filling with zeros, can give acceptable results but not better than our simultaneous algorithm.

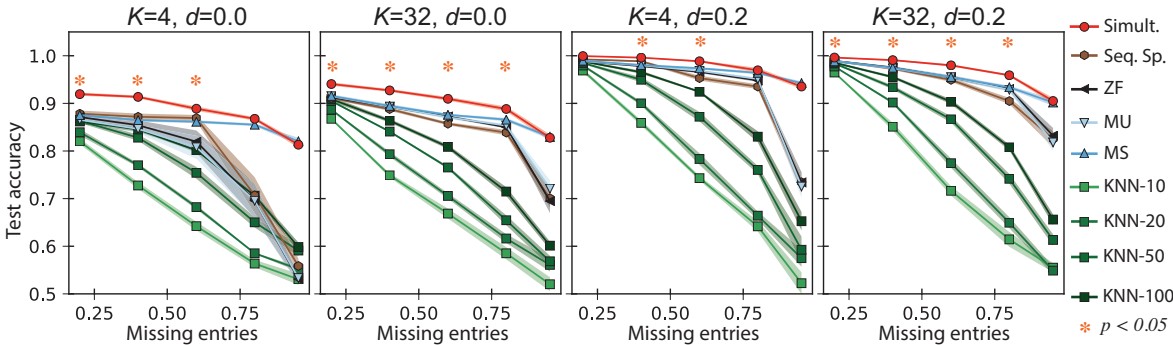

Figure 2: Experimental results on synthetic dataset using our "simultaneous" (red thick solid line) compared to various "sequential" methods. Obtained test accuracy (mean $\pm$ s.e.m computed over 10 realizations) as a function of the percentage of missing entries for separation of classes $d = 0.0, 0.2$ and levels of sparsity $K = 4, 32$. Statistical significance for the Simul-MS difference is shown ($p < 0.05$).

We also applied our algorithm to three popular computer vision datasets: MNIST (LeCun et al., 1989) and Fashion (Xiao et al., 2017) consisting of 70,000 images (60,000 train + 10,000 test) each; and CIFAR10 (Krizhevsky & Hinton, 2009) having 60,000 images (50,000 train + 10,000 test). MNIST/Fashion datasets contains $28 \times 28$ gray scale images while CIFAR10 dataset is built upon $32 \times 32 \times 3$ color images of different objects. The corresponding data sample size is $N = 28 \times 28 = 784$ for MNIST/Fashion and $N = 32 \times 32 \times 3 = 3,072$ for CIFAR10. We considered a dictionary of size $784 \times 784$ (MNIST/Fashion) and $1,024 \times 1,024$ (CIFAR10) and applied our simultaneous algorithm to learn the classifier on incomplete data for these datasets using uniform random missing masks at several levels of missing entries (25%, 50% and 75%) and for 50% random partial occlusions for MNIST/Fashion. We used a logistic regression classifier (single layer neural network) and a 4-layer convolutional neural network (CNN4) for the MNIST/Fashion dataset using batch normalization (BN) in the Fashion dataset, while for CIFAR10 dataset, a residual neural network (Resnet-18, (He et al., 2016)) was implemented. Table 1, columns $3^{rd}$ through $10^{th}$, show the accuracy obtained in the test step, when the model was trained on incomplete data and applied to incomplete as well as

| Dataset | Classifier | Random missing entries %Train / %Test | | | | | | Occlusion %Train / %Test | | Baseline %Train / %Test |
|---------|-----------|-------|-------|-------|------|------|------|-------|------|------|
| | | 75/75 | 50/50 | 25/25 | 75/0 | 50/0 | 25/0 | 50/50 | 50/0 | 0/0 |
| MNIST | Log. Reg. | 90.45 | 93.68 | 94.14 | 91.94 | 93.44 | 94.43 | - | - | 91.95 |
| | CNN4 | 94.62 | 98.34 | 98.94 | 98.14 | 98.94 | 98.95 | 88.55 | 91.37 | 98.95 |
| Fashion | CNN4+BN | 83.71 | 86.09 | 86.38 | 86.39 | 87.11 | 87.04 | 81.73 | 82.47 | 90.76 |
| CIFAR10 | Resnet18 | 43.73 | 47.27 | 47.15 | 44.07 | 45.87 | 46.45 | - | - | 86.22 |

Table 1: Test accuracy: columns $3^{rd}$ through $10^{th}$ show the results with incomplete training data and incomplete/complete test data. The baseline results were obtained by training the model on complete data.

to complete test data. The right-most column shows the baseline results obtained by training the model on complete training dataset using the implementations found in [2] and [3]. We implemented our algorithm in Pytorch 1.0.0 and ran the experiments on a single GPU. Note that for the logistic regression classifier, we obtained better results when training with incomplete data rather than using complete data. Also, it is highlighted that training on incomplete data, with 50% or less random missing entries, provides the same test accuracy as training on complete data for MNIST dataset. The hyper-parameters $\lambda_1$ and $\lambda_2$ were adjusted by cross-validation through a grid-search, as shown in Appendix C. In Fig. 3 we present some selected examples comparing the original images in the MNIST/Fashion test dataset, their observations using random masks and partial occlusions, and reconstructions computed using their sparse representation over a dictionary learned from the incomplete training data. Additional visual examples are provided in Appendix C.

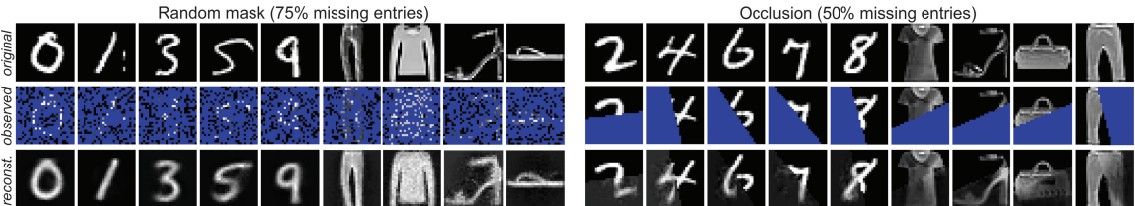

Figure 3: Reconstruction of MNIST/Fashion test data with missing entries: original images (top), incomplete observations (middle) and their reconstructions (bottom).

## 6 CONCLUSIONS

We demonstrated that assuming a sparse representation model for input data vectors allows the successful learning of a general classifier on incomplete data. We analyzed the limitations of the classical imputation approach and demonstrated through experiments that our simultaneous algorithm always outperforms sequential methods for various cases such as LASSO, zero-filling, supervised/unsupervised mean and KNN based methods. Our approach is conceptually similar to the work in Ghahramani & Jordan (1993) but, instead of using a probabilistic model, we use sparse coding, which is known to fit very well for natural signals. It is also noted that in Huang & Aviyente (2006), sparse representations of data vectors were incorporated in the optimization process for maximizing the Fisher discriminant in a context of classification. However, no theoretical guarantee was presented in that paper and our approach can be applied to any deep learning classifier architecture by simply incorporating a sparse data model, and adding two regularization terms to the objective function: one to measure the accuracy of the sparse model to explain the observations, and another to favor sparsity of the data representation, which is based on the $\ell_1$ norm.

[2] https://github.com/pytorch/examples/tree/master/mnist

[3] https://github.com/kuangliu/pytorch-cifar

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

## A  PROOF OF THEOREM 3.2

*Proof.* Let us first consider the case with $y_i = 1$. We assume that we are able to find a perfect classifier on the reconstructed full data vector such that

$$f(\hat{\mathbf{x}}_i) = \mathbf{w}_i^T \hat{\mathbf{x}}_i + b > \epsilon > 0, \tag{6}$$

where $\hat{\mathbf{x}}_i = \mathbf{D}_i' \mathbf{s}_i'$. Now, we note that the $K$-sparse representation solution given the observation is non-unique, i.e. $\exists \mathbf{s}_i, \mathbf{s}_i' \in \Sigma_K^L$, such that $\tilde{\mathbf{x}}_i = \tilde{\mathbf{D}}_i \mathbf{s}_i = \tilde{\mathbf{D}}_i' \mathbf{s}_i'$. We assume that $\mathbf{s}_i$ is the true $K$-sparse vector such that the unobserved full vector is $\mathbf{x}_i = \mathbf{D}_i \mathbf{s}_i$ and $\mathbf{s}_i'$ provides an alternative reconstruction through $\mathbf{D}_i'$ which explains the observations but $\hat{\mathbf{x}}_i \neq \mathbf{x}_i$.

Analogously to equation (1), we can find a partition of the reconstructed vector $\hat{\mathbf{x}}_i$ as follows:

$$\hat{\mathbf{x}}_i = \begin{bmatrix} \tilde{\mathbf{x}}_i \\ \mathbf{z}_i' \end{bmatrix} = \begin{bmatrix} \tilde{\mathbf{D}}_i' \\ \mathbf{E}_i' \end{bmatrix} \mathbf{s}_i', \tag{7}$$

where $\mathbf{z}_i' \in \mathbf{R}^{(N-M)}$, and we want to prove that $f(\hat{\mathbf{x}}_i) > \epsilon$ implies $f(\mathbf{x}_i) > 0$, i.e. the original data vector $\mathbf{x}_i$ (unobserved) is also correctly classified by the logistic regression with parameters $\{\mathbf{w}_i, b\}$. To that end we need to evaluate the following expression:

$$f(\mathbf{x}_i) = \mathbf{w}_i^T \mathbf{x}_i + b = \tilde{\mathbf{w}}_i^T \tilde{\mathbf{x}}_i + \mathbf{u}_i^T \mathbf{z}_i + b, \tag{8}$$

where equation (3) was used. If we add and substract $\mathbf{u}_i^T \mathbf{z}_i'$ and arrange the terms in the previous equation, we get:

$$\begin{align} f(\mathbf{x}_i) &= \tilde{\mathbf{w}}_i^T \tilde{\mathbf{x}}_i + \mathbf{u}_i^T \mathbf{z}_i' + b + \mathbf{u}_i^T (\mathbf{z}_i - \mathbf{z}_i'), \tag{9} \\ &= f(\hat{\mathbf{x}}_i) + \mathbf{u}_i^T \mathbf{E}_i \mathbf{s}_i - \mathbf{u}_i^T \mathbf{E}_i' \mathbf{s}_i'. \tag{10} \end{align}$$

The first term in the right-hand side of last equation is $f(\hat{\mathbf{x}}_i) > \epsilon > 0$ so if we show that $|\mathbf{u}_i^T \mathbf{E}_i \mathbf{s}_i - \mathbf{u}_i^T \mathbf{E}_i' \mathbf{s}_i'| < \epsilon$, then the proof will be complete. To do so, we can write:

$$|\mathbf{u}_i^T \mathbf{E}_i \mathbf{s}_i - \mathbf{u}_i^T \mathbf{E}_i' \mathbf{s}_i'| \leq |\mathbf{u}_i^T \mathbf{E}_i \mathbf{s}_i| + |\mathbf{u}_i^T \mathbf{E}_i' \mathbf{s}_i'|, \tag{11}$$

and

$$|\mathbf{u}_i^T \mathbf{E}_i \mathbf{s}_i| = \Big| \sum_{m=1}^{N-M} \mathbf{u}_i(m) \sum_{n=1}^{N} \mathbf{E}_i(m,n) \mathbf{s}_i(n) \Big| \leq \sum_{m=1}^{N-M} |\mathbf{u}_i(m)| \sum_{n=1}^{N} |\mathbf{E}_i(m,n)| |\mathbf{s}_i(n)|. \tag{12}$$

Since we assumed normalized vectors $\|\mathbf{x}_i\| \leq 1$, by applying the left-hand side of the RIP we obtain: $\|\mathbf{s}_i\| \leq 1/(1 - \delta_K)$, which implies that $|s_i(n)| \leq 1/(1 - \delta_K)$. Now, taking into account that $|\mathbf{E}_i(m,n)| \leq 1$ (columns of $\mathbf{D}$ are unit-norm) and using the fact that $\mathbf{s}_i \in \Sigma_K^L$ we obtain: $|\mathbf{u}_i^T \mathbf{E}_i \mathbf{s}_i| \leq \frac{K}{(1-\delta_K)} \sum_{m=1}^{N-M} |\mathbf{u}_i(m)|$; and, similarly, we can obtain that $|\mathbf{u}_i^T \mathbf{E}_i' \mathbf{s}_i'| \leq \frac{K}{(1-\delta_K)} \sum_{m=1}^{N-M} |\mathbf{u}_i(m)|$. Putting everything together into equation (11) we get:

$$|\mathbf{u}_i^T \mathbf{E}_i \mathbf{s}_i - \mathbf{u}_i^T \mathbf{E}_i' \mathbf{s}_i'| \leq \frac{2K}{(1 - \delta_K)} \sum_{m=1}^{N-M} |\mathbf{u}_i(m)| < \epsilon, \tag{13}$$

where we used that $\sum_{m=1}^{N-M} |\mathbf{u}_i(m)| < \frac{\epsilon(1-\delta_K)}{2K}$, which completes the proof for the case $y_i = 1$. Similarly, it is strait-forward to prove that, if $y_i = 0$, having $f(\hat{\mathbf{x}}_i) < -\epsilon < 0$ implies that $f(\mathbf{x}_i) < 0$. $\square$

# B  ALGORITHMS

Here, the pseudocode of the algorithms discussed in the paper are presented. In Algorithm 1, the simultaneous classification and coding method is described. It consists in the iterative alternation between the update of the classifier's parameters together with the dictionary, and the update of the sparse coefficients for the representation of data observations. Once the classifier is trained, we are able to apply it to incomplete test data by using Algorithm 2, where for fixed $\Theta$ and $\mathbf{D}$, we need to find the corresponding sparse coefficients $\mathbf{s}_i$, compute the full data vector estimations and, finally, apply the classifier.

The standard sparsity-based imputation method is presented in Algorithm 3 (sequential approach), which consists of learning first the optimal $\mathbf{D}$ and sparse coefficients $\mathbf{s}_i$ compatible with the incomplete observations (dictionary learning and coding phase), followed by the training phase, where the classifier is tuned in order to minimize the classification error of the reconstructed input data vectors $\hat{\mathbf{x}}_i = \mathbf{D}\mathbf{s}_i$.

---

**Algorithm 1** : Simultaneous classification and coding (training on incomplete data)

---

**Require:** Incomplete data vectors and their labels $\{\tilde{\mathbf{x}}_i, y_i\}$, $i = 1, 2, \ldots, I$, hyper-parameters $\alpha$, $\lambda_1$ and $\lambda_2$, number of iterations $N_{iter}$ and update rate $\sigma$
**Ensure:** Classifier parameters $\Theta$ and sparse representation of full data vectors $\hat{\mathbf{x}}_i = \mathbf{D}\mathbf{s}_i, \forall i$
 1: Initialize $\Theta, \mathbf{D}, \mathbf{s}_i, \forall i$ randomly
 2: **for** $n \leq N$ **do**
 3:    Fix $\mathbf{s}_i$, update $\Theta$ and $\mathbf{D}$:
 4:    $\Theta = \Theta - \sigma\frac{\partial J}{\partial \Theta}$, $\mathbf{D} = \mathbf{D} - \sigma\frac{\partial J}{\partial \mathbf{D}}$
 5:    Normalize columns of matrix $\mathbf{D}$
 6:    Fix $\Theta$ and $\mathbf{D}$, update $\mathbf{s}_i$, $\forall i$:
 7:    $\Delta_i = -\sigma\frac{\partial J}{\partial \mathbf{s}_i}$, $\forall i$
 8:    **if** $\mathbf{s}_i(j)[\mathbf{s}_i(j) + \Delta_i(j)] < 0$ **then**
 9:       $\Delta_i(j) = -\mathbf{s}_i(j)$; avoid zero crossing
10:    **end if**
11:    $\mathbf{s}_i = \mathbf{s}_i + \Delta_i$, $\forall i$
12: **end for**
13: **return** $\Theta, \mathbf{D}, \mathbf{s}_i, \hat{\mathbf{x}}_i = \mathbf{D}\mathbf{s}_i, \forall i$

---

**Algorithm 2** : Testing on incomplete data

---

**Require:** Incomplete data vectors $\{\tilde{\mathbf{x}}_i\}$, $i = 1, 2, \ldots, I$, classifier parameters $\Theta$, dictionary $\mathbf{D}$, hyper-parameters $\lambda_1$ and $\lambda_2$, number of iterations $N_{iter}$ and update rate $\sigma$
**Ensure:** Class assigned to each vector $y_i$ and sparse representation of full data vectors $\hat{\mathbf{x}}_i = \mathbf{D}\mathbf{s}_i, \forall i$
 1: **Sparse coding stage:** for fixed dictionary $\mathbf{D}$ find sparse representations of observations $\tilde{\mathbf{x}}_i$
 2: Initialize $\mathbf{s}_i, \forall i$ randomly
 3: **for** $n \leq N_{iter}$ **do**
 4:    $\Delta_i = -\sigma\left[\lambda_1\frac{\partial J_1}{\partial \mathbf{s}_i} + \lambda_2\frac{\partial J_2}{\partial \mathbf{s}_i}\right]$, $\forall i$
 5:    **if** $\mathbf{s}_i(j)[\mathbf{s}_i(j) + \Delta_i(j)] < 0$ **then**
 6:       $\Delta_i(j) = -\mathbf{s}_i(j)$; avoid zero crossing
 7:    **end if**
 8:    $\mathbf{s}_i = \mathbf{s}_i - \sigma\Delta_i$, $\forall i$
 9: **end for**
10: $\hat{\mathbf{x}}_i = \mathbf{D}\mathbf{s}_i, \forall i$; Compute reconstructions of unobserved vector data $\mathbf{x}_i$
11: **Classification stage:** apply classifier to reconstructions $\mathbf{x}_i$
12: $y_i = \arg\max_y(p_\Theta^y(\hat{\mathbf{x}}))$
13: **return** $\Theta, y_i, \mathbf{s}_i, \hat{\mathbf{x}}_i, \forall i$

---

---

**Algorithm 3** : Sequential approach (imputation method)

---

**Require:** Incomplete data vectors and their labels $\{\tilde{\mathbf{x}}_i, y_i\}$, $i = 1, 2, \ldots, I$, hyper-parameters $\alpha$, $\lambda_1$ and $\lambda_2$, number of iterations $N_{iter}$ and update rate $\sigma$

**Ensure:** Classifier parameters $\Theta$ and sparse representation of full data vectors $\hat{\mathbf{x}}_i = \mathbf{D}\mathbf{s}_i, \forall i$

 1: Initialize $\mathbf{D}, \mathbf{s}_i, \forall i$ randomly
 2: `Dictionary learning and coding stage:` update $\mathbf{D}$ and $\mathbf{s}_i$
 3: **for** $n \leq N$ **do**
 4:    $\mathbf{D} = \mathbf{D} - \sigma \frac{\partial J_1}{\partial \mathbf{D}}$
 5:    `Normalize columns of matrix` $\mathbf{D}$
 6:    $\Delta_i = -\sigma \left[ \lambda_1 \frac{\partial J_1}{\partial \mathbf{s}_i} + \lambda_2 \frac{\partial J_2}{\partial \mathbf{s}_i} \right], \forall i$
 7:    **if** $\mathbf{s}_i(j)[\mathbf{s}_i(j) + \Delta_i(j)] < 0$ **then**
 8:      $\Delta_i(j) = -\mathbf{s}_i(j)$; `avoid zero crossing`
 9:    **end if**
10:    $\mathbf{s}_i = \mathbf{s}_i + \Delta_i, \forall i$
11: **end for**
12: $\hat{\mathbf{x}}_i = \mathbf{D}\mathbf{s}_i, \forall i$; Compute reconstructions of unobserved vector data $\mathbf{x}_i$
13: `Training stage:` update $\Theta$
14: **for** $n \leq N$ **do**
15:    $\Theta = \Theta - \sigma \frac{\partial J_0}{\partial \Theta}$; `Update` $\Theta$:
16: **end for**
17: **return** $\Theta, \mathbf{D}, \mathbf{s}_i, \hat{\mathbf{x}}_i = \mathbf{D}\mathbf{s}_i, \forall i$

---

## C ADDITIONAL EXPERIMENTAL RESULTS

### C.1 COMPARISON TO SEQUENTIAL METHODS ON BENCHMARK DATASETS

In Table 2, we compare our simultaneous algorithm (Simul) with the following standard sequential methods: Zero Fill - ZF, Mean Supervised - MS, KNN-10, KNN-20, KNN-50 and KNN-100. We computed the Test Accuracy on incomplete data using random masks for MNIST and CIFAR10 datasets. It is noted that the bests result using standard sequential method were obtained with KNN-10 or KNN-20. However, the simultaneous algorithm outperforms KNN in all the cases increasing the performance by approximately 10.% for MNIST and CIFAR10, when the percentage of random missing features is 75%.

### C.2 HYPERPARAMETER TUNNING

In Table 3 and Figure 4 we present the results of the grid search for hyper-parameter tuning on MNIST and CIFAR10 datasets. We fit our model to the training dataset for a range of values of parameters $\lambda_1$ and $\lambda_2$ and apply it to the test data set. Figure 4 shows the accuracy obtained for the test dataset with different classifiers and levels of missing entries. We chose the hyper-parameters values such that the test accuracy is maximum, as shown in Table 3.

### C.3 ADDITIONAL VISUAL RESULTS

To visually evaluate our results, additional selected examples of original (complete) images of the test dataset in MNIST and Fashion, together with their given incomplete observations and obtained reconstructions, are shown in Figure 5 and Figure 6

| MNIST (CNN4) | | | | | | | |
|---|---|---|---|---|---|---|---|
| **Missing** | **ZF** | **MS** | **KNN10** | **KNN20** | **KNN50** | **KNN100** | **Simul.** |
| **75%** | 84.14 | 83.69 | **88.10** | 87.68 | 86.77 | 86.11 | **98.14** |
| **50%** | 89.65 | 88.40 | **91.22** | 90.94 | 90.72 | 90.68 | **98.94** |
| CIFAR10 (Resnet18) | | | | | | | |
| **Missing** | **ZF** | **MS** | **KNN10** | **KNN20** | **KNN50** | **KNN100** | **Simul.** |
| **75%** | 18.7 | 21.98 | 33.10 | **34.78** | 29.56 | 30.55 | **44.07** |
| **50%** | 41.35 | 17.79 | 31.54 | **36.94** | 31.16 | 30.78 | **45.87** |

Table 2: Test Accuracy obtained on MNIST and CIFAR10 dataset using standard sequential methods and compared to our simultaneous algorithm.

| Dataset | Classifier | Random missing entries | | | | | | Occlusion | |
|---|---|---|---|---|---|---|---|---|---|
| | | **75%** | | **50%** | | **25%** | | **50%** | |
| | | $\lambda_1$ | $\lambda_2$ | $\lambda_1$ | $\lambda_2$ | $\lambda_1$ | $\lambda_2$ | $\lambda_1$ | $\lambda_2$ |
| **MNIST** | **Log. Reg.** | 0.32 | 1.28 | 0.64 | 1.28 | 0.64 | 1.28 | - | - |
| | **CNN4** | 1.28 | 1.28 | 2.56 | 1.28 | 5.12 | 1.28 | 10.24 | 10.24 |
| **CIFAR10** | **Resnet18** | 0.024 | 0.008 | 0.032 | 0.004 | 0.032 | 0.01 | - | - |

Table 3: Hyper-parameter tuning: crossvalidated hyperparameters $\lambda_1$ and $\lambda_2$ obtained for MNIST and CIFAR10 datasets with the classifiers used in our experiments.

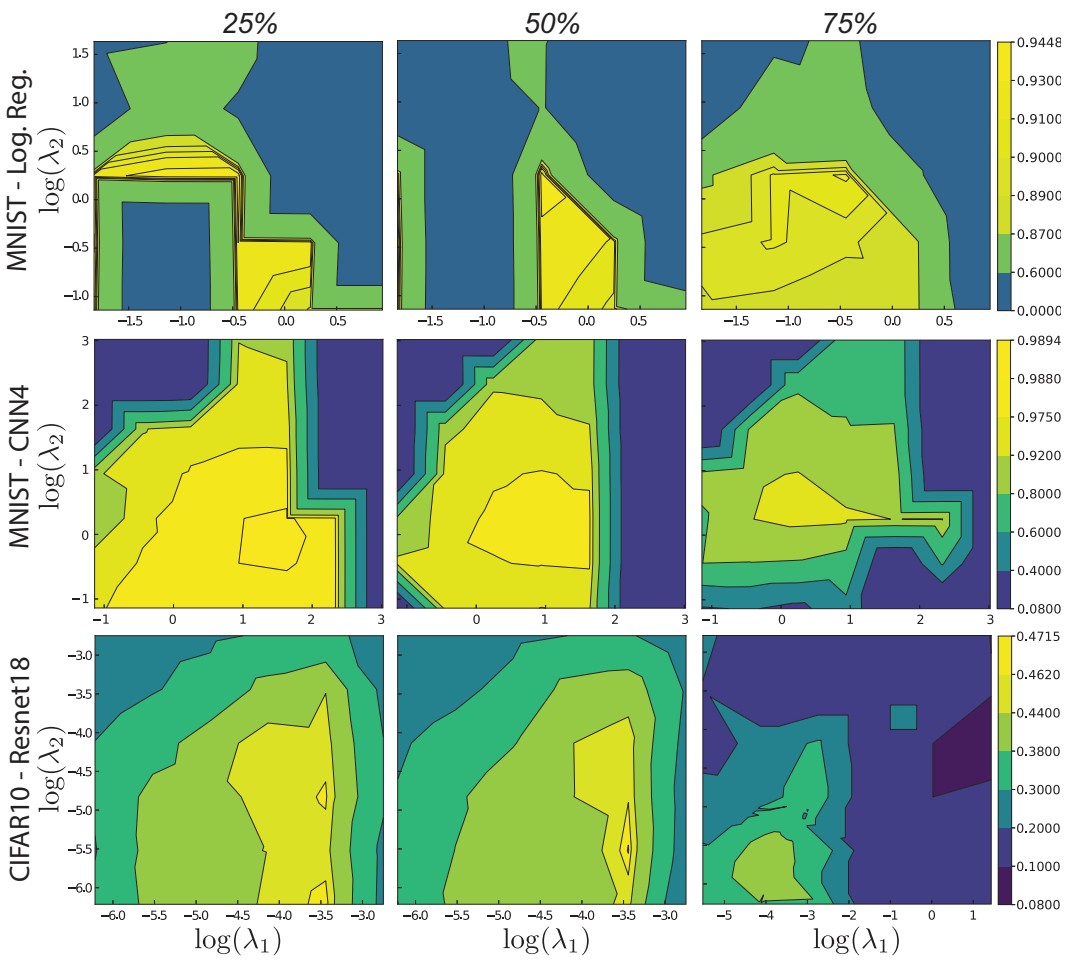

Figure 4: Testing accuracy in the grid search for hyper-parameter tuning: $\lambda_1$ and $\lambda_2$ were chosen through crossvalidation by maximizing the testing accuracy in all datasets with different levels of missing entries: 25%, 50% and 75%.

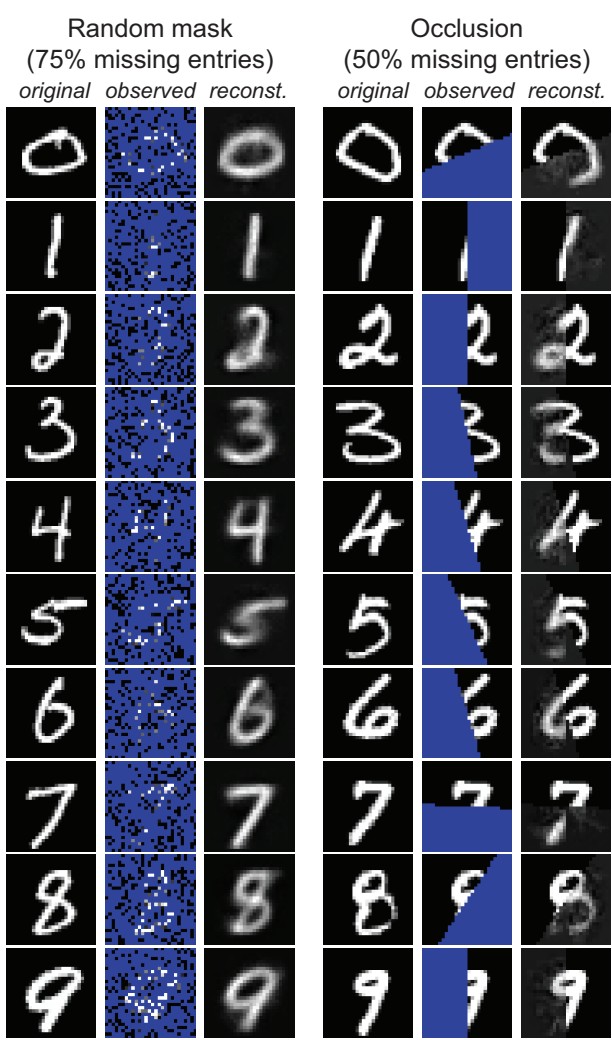

Figure 5: Reconstructions of incomplete test MNIST dataset vectors by applying our simultaneous classification and coding algorithm using the CNN4 architecture.

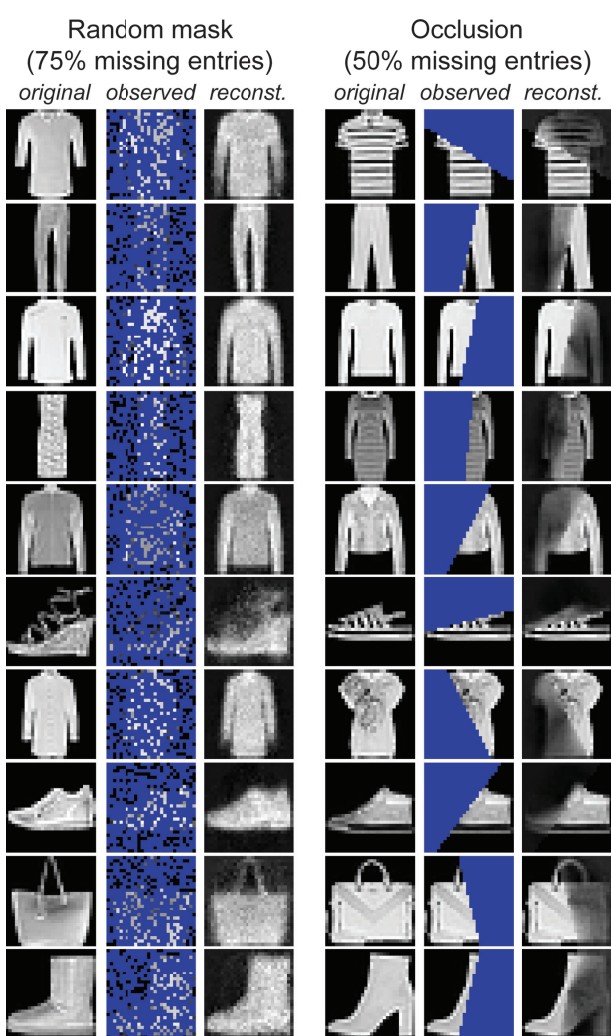

Figure 6: Reconstructions of incomplete test Fashion dataset vectors by applying our simultaneous classification and coding algorithm using the CNN4 architecture with Batch Normalization (BN).

