# OpenReview forum: "Supervised learning with incomplete data via sparse representations"
_ICLR.cc/2020/Conference — Reject_

### Official Review · AnonReviewer3 · 2019-10-07
**Official Blind Review #3**

**Rating:** 6

**Review:**

1. Summary
The authors propose a scheme for simultaneous dictionary learning and classification based on sparse representation of the data within the learned dictionary. Their goal is achieved via optimization of a three-part training cost function that explicitly models the accuracy and sparsity of the sparse model, simultaneously with usual classification error minimization. An alternating optimization algorithm is used, alternating between sparse representations and other parameters.

The problem they want to address with this scheme is training over incomplete/partial feature vectors. A clean theoretical statement is provided that provides conditions under which a classifier trained via partial feature vectors would do no better in terms of accuracy, had it been trained on complete feature vectors. The authors claim this condition can be checked after training, although this ability is not validated/illustrated numerically.

2. Decision and Arguments
Weak Accept
a) Very nice mathematical result and justification. The proof is clear. However, why wasn’t the result used in the numerical section? You claim that the condition (4) can be evaluated to test optimality: how do your trained dictionaries compare to say another dictionary learning scheme? After all, this doesn’t seem to inform your actual learning scheme and is not used to evaluate your results. It would be nice to see a numerical validation/illustration of this result. Also is \delta_K really that easy to compute?
b) The numerical results are good—but lack error bars and comparators. I don’t understand why you considered so many comparators on synthetic data and none on more ‘realistic’ benchmark data.
c) Also I don't feel very satisfied doing image examples, it would be more interesting to work on difficult (eg medical) classification problems with large feature vectors

4. Additional Feedback
a) Very well and clearly written with intuitive examples and clean math. My only suggestion is to clarify in the abstract that there are missing *features*-- when I first read "incomplete data" I think of entire data samples that are missing from the training set. That makes no sense, but it became clear when I got to section 2.
b) Please use markers and dashes etc. with *every* line in your plots. It is hard (or impossible for many) to compare as is with such tiny thin lines.

5. Questions
a) Could you comment on statistical significance of your results? For synthetic data it should be easy to perform the experiments on a number of mask realizations and include error bars.
b) Why no comparators on benchmark datasets?
c) The proof of Thm 3.2 is nice—but how reasonable is the assumption that you have two dictionaries each with the exact same RIP constant \delta_K? Can that property be enforced (even approximately) during training? Or is it trivial that, given one such dictionary, there exists a second one?
d) See 2.a


**Experience Assessment:**

I have published one or two papers in this area.

**Review Assessment: Checking Correctness Of Derivations And Theory:**

I carefully checked the derivations and theory.

**Review Assessment: Checking Correctness Of Experiments:**

I carefully checked the experiments.

**Review Assessment: Thoroughness In Paper Reading:**

I read the paper thoroughly.

---

> ### Author Response · Authors · 2019-11-14
> **New revised version with improved experimental results based on your critical and constructive comments**
>
> Thanks for the detailed and highly constructive review. We have considered all the comments and used them to improve the paper by including new results. Please check, our detailed responses below:
> a) Thanks for the positive comment. The theorem states a sufficient condition for optimal learning. Unfortunately, it is well known that RIP constant is difficult to compute in practice [1] and that is why we haven’t computed it in our experiments. However, we would like to mention that, when the dictionary is highly uncorrelated, i.e. if \rho K < 1, then the RIP constant can be written in terms of the correlation coefficients \rho as follows: \delta_K =(K-1)\rho [1], and in this case, \rho is easy to compute. Unfortunately, this highly uncorrelated dictionary condition was verified few times during our simulations. To give an idea, obtained correlation coefficients of learned dictionaries were around \rho = 0.35, which was in the same order of the correlation coefficient of the true dictionary used for generating the data. Please, note that, to replace the RIP constant by its computation based on \rho, we should have K < 1/0.34 = 2.85, which means that we should have our data set represented by only K=2 atoms in the dictionary which is not realistic.
> b) We agree with the reviewer and thank for this important comment.
> We added experimental results on real datasets comparing our method against meaningful baseline methods (sequential methods). See Table 2 in the revised version of the paper. Please note that the best result using standard sequential methods was obtained with KNN-10 or KNN-20. However, our algorithm outperforms KNN in all cases, for example, increasing the performance by approximately 10% for MNIST and CIFAR10 with a 75% percentage of random missing features.
> We also added error bars and a rigorous statistical significance analysis of our experimental results on synthetically generated data (see response to question (a) below)
> c) The purpose of using these vision datasets was to provide a proof of concept for our method, especially because we can look on reconstructed images and visually evaluate how the model is adapted to the underlying classes (see Fig. 3 for example). We totally agree that the potential of this method is to be applied to more difficult classification problems with large feature vectors. In fact, we already are working on applying this strategy for Brain Computer Interface problems where, based on raw EEG signals, we need to interpret the user intention. Unfortunately, we are not able to include our results in this paper but we expect we will share them to the community soon.
>
> Response to Additional Feedback a): We introduced the term “missing features” in the new abstract in order to avoid confusion. We also modified Fig. 2 by not only adding error bars and statistical significance but also adding markers as suggested. We think the figures is greatly improved now.
> Responses to specific questions:
> a) In the revised version, we included results obtained by computing the accuracies over 10 realizations for each case (sparsity K, separation \mu, missing percentage). Please, note that for each independent realization we used a different random mask and new generated data samples according to the sparse model. As a result, we report in the new Fig. 2, the plots of the mean +/- s.e.m (standard error of the mean). We also performed a t-test (p=0.05) to evaluate the statistical significance of the difference between our algorithm and the Mean Supervised (MS) classical method, which is the one that performed the best most of the times. The statistically significant results are indicated by an asterisk (*) in the same figure. We highlight, that our algorithm provided very stable results (low variance) compared to other classical methods.
> b) We apologize for not including comparators on benchmark datasets in the original submission. In the revised version, we included in Table 2, a comparison of classical sequential methods with our algorithm for MNIST and CIFAR10 datasets.
> c) In general, the solution is non-unique, and it is possible that another Dictionary could serve as solution. Please note that if a dictionary has RIP constant \delta_K, then it has also RIP constant \delta’_K if \delta’_K>\delta_K, therefore we can choose the maximum value to define a common RIP constant.
> We agree that having smaller \delta_K would improve the quality of the learned classifier, thus it would be good idea to incorporate somehow a mechanism during the optimization in order to enforce this property, for example, by minimizing correlation between dictionary columns. However, we haven’t tried it yet because the obtained results seemed to be satisfactory enough. We think this would be a good idea for a future direction of the work.
> d) See response to point 2a
> References:
> [1] YC Eldar and Gitta Kutyniok. Compressed Sensing: Theory and Applications. New York: Cambridge Univ. Press, 20:12, 2012.

---

### Official Review · AnonReviewer2 · 2019-10-24
**Official Blind Review #2**

**Rating:** 6

**Review:**

This manuscript studies supervised learning with incomplete observation of the features, assuming a low-rank structure in the full set of features. The work tackles the problem with a global cost function that optimizes the classifier on observations reconstructed with a dictionary penalized jointly for sparse coding. Importantly, the dictionary is optimized both for the supervised-learning task (the classifier) and the sparse coding. The manuscript contributes a theoretical argument showing with sparse-recovery arguments that if the data can be decomposed sparsely, the incomplete feature prediction problem can perform as well as the complete one. Finally, it performs a empirical study on simulated data as well as computer-vision problems with random missing pixels as well as occlusions, and using for the classifier logistic regression as well as classic neural network architectures.

The work is interesting, but I have several big-picture comments on the framing of the work. The first one is that framing the problem as a missing values one, referring to the classic missing-values literature, does not seem right to me. Indeed, in the missing-values literature, different features are observed on different samples, which leads to a set of problems and corresponding solutions. The contribution here is a different one. My second big picture comment is that it seems that this is very much related to the classic work "supervised dictionary learning" by Mairal et al. The framing of the Mairal paper is that of a representation-learning architecture: the cost function is there to impose an inductive bias. Added to a linear model, it contributes representation learning, and hence can bring some of the appealing properties of deep architectures. This is very much what the experimental study reveals, where the contributed architecture with latent factors outpeforms the linear model on fully observed data. With these two comments, I would like to know better how the work positions itself to supervised dictionary learning: the paper is cited (albeit with an incorrect citation), but the positioning is not explicit.

**Experience Assessment:**

I have published in this field for several years.

**Review Assessment: Checking Correctness Of Derivations And Theory:**

I assessed the sensibility of the derivations and theory.

**Review Assessment: Checking Correctness Of Experiments:**

I assessed the sensibility of the experiments.

**Review Assessment: Thoroughness In Paper Reading:**

I read the paper thoroughly.

---

> ### Author Response · Authors · 2019-11-14
> **Problem setting was misunderstood and relation to Mairal et al (2008) work**
>
> Thank you for your time and expertise put into the review. Please, check our responses because we believe there was a misunderstanding about the problem setting, which is now clarified in the revision.
> 1- We think that the Reviewer has misunderstood our problem setting. In our case, and as in the classical missing-values literature, the pattern of missing entries (random or structured) is different for each sample and there is no “low-rank structure in the full set of features” as the Reviewer assumed. Maybe, the confusion was caused by the fact that in our mathematical notation we shuffled the entries of samples (and rows of dictionary) in order to have the observed values placed in the upper M positions. Then, for each sample, we have a sparse representation given by x_i = D_i s_i, where D_i is a permuted version of the global dictionary D learned by our algorithm. We have added a new statement in the paper in order to avoid such a confusion (see abstract and page 2 and page 6).
> 2- We agree that our work is somehow related to [1], however there are some important differences we would like to point out:
> (i) Paper [1] does not consider missing entries. They assumed a complete dataset.
> (ii) In [1], the term in the cost function associated to the loss of classification is based on the application of the softmax function to two simple proposed forms: a) Linear on the sparse vectors of coefficients s_i or b) Bilinear on vector s_i and data sample x_i. These two functions have not an explicit dependence on the dictionary D. In our work, instead, when we consider the logistic regression classifier, the corresponding function is linear on the reconstructed version of the data sample \hat{x}_i = Ds_i, thus the function is still linear on the vector s_i as in [1], but explicitly dependent on the dictionary D. Moreover, our approach allows us to use the extraordinary classification power of neural networks based on the reconstructions, which allows to learn a dictionary D that not only allows for a sparse and accurate representation but also provides a good classification of reconstructions.
> We think it is difficult to say how our method positions itself to supervised dictionary learning because both methods have similarities but are not equivalent and solve different problems.
> We thank the Reviewer to highlight our experimental results showing that a linear model based on incomplete data outperforms the linear model on fully observed data. This is a very interesting fact that we would like to better understand in a future work. Our interpretation of this is that the linear model (Log. Reg.) is too simple for complex data distributions (fully observed data), however we can improve the classification performance using the same simple linear model but applied to transformed data samples (reconstructions) based on a limited but meaningful subset of entries.
>
> We thank the reviewer for pointing out that our previous citation to Mairal et al was incorrect. We fixed it in the revision.
>
> References
> [1] Julien Mairal, Jean Ponce, Guillermo Sapiro, Andrew Zisserman, and Francis R Bach. Supervised Dictionary Learning. In Advances in Neural Information Processing Systems (NIPS), 2008.

---

> > ### Comment · AnonReviewer2 · 2019-11-15
> > **This is supervised imputation, but the formalism of the paper is incorrect**
> >
> > I am not very satisfied with the points 1 that the authors make in their reply. Indeed, if the entries are not the same in each observation, there is a missing random permutation matrix from the formalism. Studying the manuscript further, I realize that it is inconsistent, as in the paragraph below equation 5, the authors add a mask to account for missing values in the loss, and do not permute the data.
> >
> > Beyond these formalism points (which I find make the paper hard to read), I know realize that the paper is about supervised imputation. I find that this is an interesting point. However the theoretical results hold in restricted settings, and the experiments only study cases that are quite unusual in the missing-data literature: dropped pixels in images, rather than missing information in questionnaires.
> >
> > Nevertheless, I have raised my rating.

---

### Official Review · AnonReviewer4 · 2019-10-29
**Official Blind Review #4**

**Rating:** 1

**Review:**

This paper proposes a method for simultaneously imputing data and training a classifier. Here are my detailed comments:

1. The proposed method is a simple combination of dictionary learning and classification. As can be seen from (5), J_1 and J_2 are the loss function for dictionary learning, and J_0 is the empirical risk for classification (J_0). I did not see much novelty here.

2. There exists a capacity mismatch between data imputation and classifier. The paper claims that the method is proposed for training deep neural networks. However, deep neural networks are capable of modeling very complex distributions (e.g., images), and the dictionary learning only considers a very simple bilinear factorization, which is not of the same modeling capability as neural networks.

3. The experimental evaluations are very weak. Table 1 even does not provide any meaningful baseline methods, e.g., the so-called "sequential method".

4. Theorem 3.2 requires the dictionary to be RIP, which is a very strong assumption and unlikely to hold in practice, especially when considering D is actually trained by minimizing (5). The proof of Theorem 3.2 is very elementary, ad-hoc, and does not provide any insight to the problem.

**Experience Assessment:**

I have published in this field for several years.

**Review Assessment: Checking Correctness Of Derivations And Theory:**

I carefully checked the derivations and theory.

**Review Assessment: Checking Correctness Of Experiments:**

I carefully checked the experiments.

**Review Assessment: Thoroughness In Paper Reading:**

I read the paper at least twice and used my best judgement in assessing the paper.

---

> ### Author Response · Authors · 2019-11-14
> **Please check our improved revised version including stronger new results and responses to your comments**
>
> Thank you for reviewing the paper and providing critical comments. We took into consideration all the comments and provide an improved version of the paper clarifying the raised issues and including new and stronger experimental results.
> 1- Our main contribution is to shed light on the practical usefulness and the theory behind this simple method, which has not been proposed or studied before at the best of our knowledge except for the works already cited in our paper.
> 2- We would like to note the reviewer that sparse representations do have the power to accurately model very complex distributions, such as natural signals (mages, audio, EEG, etc). In fact, sparse representations based, for example, on the Discrete Cosine Transform inspired mathematicians to develop the theory of Wavelets that allows to model multidimensional natural signals almost perfectly [1].
> Additionally, in our method we rely on the extraordinary classification power of neural networks that, combined to a model based on sparse representation of data, provides a useful solution for dealing with missing entries/features as our experimental and theoretical studies illustrates.
> 3- Thanks for this comment. We added experimental results comparing our method against meaningful baseline methods (sequential methods). See Table 2 in the revised version of the paper. Please note that the best result using standard sequential methods was obtained with KNN-10 or KNN-20. However, our algorithm outperforms KNN in all cases, for example, increasing the performance by approximately 10% for MNIST and CIFAR10 with a 75% percentage of random missing features.
> Please, also note that we improved the results on synthetically generated data by including error bars and performing a rigorous statistical significance analysis of the results (see Fig. 2 and text in the revised version).
> 4- Please, note that simple orthogonal dictionaries such as Wavelet or DCT has RIP constant delta equal to zero providing optimal sparse representations in practice. It is true that we are not enforcing an optimal RIP during learning, as the reviewer noted, but our experiments showed that good results are obtained in practice. We think that by enforcing low coherence (small delta) during learning could improve the properties of the dictionary but, at the same time, reduce the power of representation (larger approximation errors). However, this could be investigated in a follow-up work.
> We also note that in the case of having a very low correlated dictionary, then the RIP constant can be written in terms of the correlation coefficient, thus providing a direct computation of the condition (4).
> We believe that our theorem provides useful insights to the problem. It provides a sufficient condition that makes the classifier obtained from incomplete data a proper one. It helps us to understand that the following conditions are desirable:
> (a) very well separated reconstructed vectors (large epsilon);
> (b) very sparse model (small K);
> (c) small norm variability of transformed K-sparse vectors through D (small delta), which can be interpreted as a quasi-orthonormal basis behavior; and
> (d) small weights assigned to unobserved entries of xi (small DMI \mu_i).
> We may agree that the theory here presented is not complete, but we believe is fundamental as it may allow future advances.
>
> References:
> [1] Mallat, Stéphane: A wavelet tour of signal processing (2. ed.). : Academic Press, 1999. - ISBN 978-0-12-466606-1

---

### Decision · Program_Chairs · 2019-12-19

**Decision:**

Reject

**Comment:**

This was a difficult paper to decide, given the strong disagreement between reviewer assessments.  After the discussion it became clear that the paper tackles some well studied issues while neglecting to cite some relevant works.  The significance and novelty of the contribution was directly challenged, yet I could not see a convincing case presented to mitigate these criticisms.  The paper needs to do a better job of placing the work in the context of the existing literature, and establishing the significance and novelty of its main contributions.